# Knowledge of Dental Trauma and Orthodontic Management of Traumatized Teeth by a Group of Lithuanian Orthodontists

**DOI:** 10.3390/medicina59071289

**Published:** 2023-07-13

**Authors:** Simona Stučinskaitė, Paulina Laugalė, Dominyka Grinkevičienė, Rita Vėberienė, Dalia Smailienė

**Affiliations:** 1Department of Orthodontics, Faculty of Odontology, Lithuanian University of Health Sciences, Lukšos-Daumanto 6, LT-50161 Kaunas, Lithuania; simona.stucinskaite1@gmail.com (S.S.); dominyka.narb@gmail.com (D.G.); dalia.smailiene@lsmu.lt (D.S.); 2Department of Dental and Oral Pathology, Lithuanian University of Health Sciences, Eiveniu 2, LT-50161 Kaunas, Lithuania; paulina.grinkeviciute@gmail.com

**Keywords:** dental trauma, orthodontic management, knowledge, survey

## Abstract

*Background and Objectives*. There is a deficiency of research on orthodontic movements and management strategies for traumatized teeth to avoid further treatment complications. The aim of the study was to evaluate the knowledge of Lithuanian orthodontists about dental trauma and the orthodontic management of traumatized teeth. *Materials and Methods*. Lithuanian orthodontists and orthodontic residents were introduced to the purpose, objectives and questionnaire of the study. The questionnaire was developed based on demographics, the participants’ personal experience, specific questions about dental traumas and orthodontic treatment tactics for patients with a history of dental trauma. *Results*. A total of 54 participants (42 orthodontists and 12 orthodontic residents) completed the survey. The overall response rate was 42.9%. The results of the survey revealed that the largest share of the study participants—61%—stated that they had treated traumatized teeth just a few times during the whole practice. Briefly, 53.7% of respondents inquired about the dental trauma history only if they saw signs of complications. The results of the survey revealed that the frequency of correct answers to questions about dental trauma was 63.7%, and that about the orthodontic treatment of traumatized teeth was 54.9%. It is worth noting that one-third of physicians pointed out the lack of information about the orthodontic management of traumatized teeth. *Conclusion.* The knowledge of Lithuanian orthodontists and orthodontic residents about dental injuries and the orthodontic treatment of traumatized teeth is insufficient. Further educational training is recommended.

## 1. Introduction

Dental trauma has a negative impact on the lives of children and adolescents due to its physical, physiological, social, economic consequences and high prevalence [1]. Dental trauma may be related to the anatomical features of the individual, such as an increased overjet (>3 mm), insufficient lip closure, and anterior open bite [1,2,3]. It has also been observed that patients with Class II Division 1 malocclusion are more likely to have dental injury than children with other types of occlusion are [2,4]. Risk factors such as not wearing mouthguards during contact sports, permanent teeth decay, obesity, and tongue piercings can lead to dental trauma [1,3].

Epidemiological studies show that most injured teeth remain untreated [3]. Scientific publications state that one in ten patients have had dental trauma prior to orthodontic treatment [5,6,7,8]. Traumatized teeth are more susceptible to complications during orthodontic treatment [5]. Possible complications include external inflammatory root resorption, cervical and internal resorptions, pulp necrosis, and root canal obliteration [5,7,9,10,11]. The complications during orthodontic tooth movement increase depending on whether or not the pulp was damaged during the trauma, and whether endodontic treatment was performed or not. If the root of the injured tooth has been affected by resorption, orthodontic tooth movement will enhance the resorption process [12]. In case reports, it was observed that previously successful regenerative cases ended unsuccessfully. Such complications as external invasive cervical resorption, regenerative tissue necrosis, or internal resorption occurred after the application of orthodontic forces [13]. These consequences can have a negative impact on an individual’s self-confidence, aesthetics, quality of life, psychosocial behavior, social acceptance and self-esteem, so they cannot be ignored [1,5]. It is important to pay extra attention to traumatized teeth during orthodontic treatment and evaluate the risk of possible complications [5]. Traumatic dental injuries could complicate and/or delay or completely alter orthodontic treatment. Traumatized teeth require a follow-up period before orthodontic treatment because the periodontal ligament must be completely healed [3,7]. The observation period is different for each injury and depends on its severity [3]. Other studies have shown insufficient knowledge about the orthodontic treatment of traumatized teeth—the frequency of correct answers is only 61.3% [14] and 36.5% [7]. Scientific publications indicate a lack of information on dental and periodontal injuries [5,14]; it is therefore important to examine this topic in more detail and to draw to the attention of existing and future orthodontists to dental trauma and their orthodontic treatment. The aim of the study was to evaluate the knowledge of Lithuanian orthodontists about dental trauma and the orthodontic management of traumatized teeth.

## 2. Materials and Methods

The survey was conducted from 1 November 2021 to 24 January 2022. The research was approved by the Kaunas Regional Biomedical Research Ethics Committee (ID number of ethical approval: BE-2-26). Lithuanian orthodontists and orthodontic residents were invited to take part in the study. Participants were introduced to the purpose, objectives and questionnaire of the study. Anonymous questionnaires were distributed during orthodontic conferences and a link to the questionnaire was emailed to 73 dental clinics where the orthodontists worked.

Inclusion criteria in the study included Lithuanian orthodontists or orthodontic residents, of any age, and in any workplace.

Exclusion criteria included general dentists, endodontists, restorative and pediatric dentists, oral surgeons and students of these subjects.

According to the data available from the state accreditation service for health care activities, there are 114 valid licenses of orthodontists in Lithuania, and 12 residents of orthodontics who are studying at Vilnius University and Kaunas Lithuanian University of Health Sciences. In total, there were 126 physicians. Data were collected in 2021.

The questionnaire was developed based on three published studies [5,7,14]. Parts of the questionnaire included the following. 

The first demographic part of the questionnaire consisted of 3 questions to examine the academic and professional profile of the respondent (work experience, and workplace).The second part (3 questions) aimed to find out about the participants’ personal experience with patients who had experienced dental trauma.The third part presented 5 specific questions about dental trauma (clinical symptoms, the most common age of patients with dental trauma, and the action required immediately after the injury).The fourth part consisted of 15 questions regarding the orthodontic management tactics for patients with a history of dental trauma (when the treatment can be started in the case of a certain trauma; complications to be expected). The last question was asked to address the reasons why respondents were unsure of when to start orthodontic treatment after dental trauma.

A pilot study was conducted before interviewing the participants. Questionnaires were submitted for completion to 10 orthodontists to assess the relevance, composition, and quality of the questions. All comments and suggestions were requested to be shared. After receiving them, the questionnaire was further improved so that the questions were precise and plain. The results of this study were not used in the analysis of the results of the final study.

During the analysis of the survey data, we decided to evaluate only the correct answers of the respondents. Correct answers were based on the guidelines for dental trauma developed by the International Association of Dental Traumatology [15,16]. Recommendations for the orthodontic treatment of dental injuries were based on articles published by Kindelan et al. [3] and Sandler et al. [6].

Statistical data analysis was coded and performed using IBM SPSS version 27.0.

In the descriptive analysis, the number of correct answers was presented as a percentage (%)indicating the exact number of respondents (*n*).

## 3. Results

A total of 54 participants (42 orthodontists and 12 orthodontic residents) completed the survey. The overall response rate was 42.9%. Work experience was from 1 to 33 years, and the median was 7.5 (2.9–15.0) years. Most of the participants worked in a private dental clinic (*n* = 48, 89%), followed by university clinics (*n* = 23, 43%) and public clinics (*n* = 5, 9%).

The frequency of correct answers to the survey was 57.1% for practicing orthodontists and 58.3% for orthodontic residents (total 57.4%). There were no significant statistical differences between these two groups of specialists; therefore, the data were not further analyzed and described in detail.

The results of the survey revealed that the largest share of the study participants—61%—stated that they had treated traumatized teeth just a few times during the whole practice and only 6% of physicians reported not seeing any patients with dental trauma (Figure 1). Assessing the way doctors learn about a patient’s past dental trauma, it was found that the majority of respondents (53.7%) said they ask about a patient’s dental trauma history only if they see signs of complications (Table 1).

The part of the survey about dental trauma revealed that the frequency of correct answers about dental trauma was 63.7% (Table 2). It was found that the frequency of correct answers to the questions about the orthodontic treatment of traumatized teeth was 54.9% (Table 3).

Analyzing the data on the orthodontic treatment of a post-traumatic ankylosed tooth, it was observed that most of the study participants chose the correct response options—the tooth will not be moved (61.1%) and the tooth can be used as an anchorage (77.8%). Both correct answers were chosen by 42.6% of the physicians.

The study also assessed the reasons why respondents are unsure of when to start the orthodontic treatment of traumatized teeth (Figure 2). Doctors stated that the orthodontic treatment of traumatized teeth should be evaluated by specialized interdisciplinary teams and that there is a lack of information about the orthodontic management of traumatized teeth.

## 4. Discussion

Kindelan et al. [3] published a review of dental trauma and the recommendations for orthodontic treatment, which is considered one of the best guidelines for the treatment of such cases [14]. The intention of this study was to draw attention to the topic of dental trauma among Lithuanian orthodontists. The survey must have consisted of specific questions about dental trauma and orthodontic treatment of traumatized teeth to be able to evaluate the orthodontists’ knowledge in a precise way. There are a few similar studies that have been conducted regarding the topic of dental trauma and th eorthodontic management of traumatized teeth. The study published by Van Gorp et al. [5] questioned three different groups of professionals: general dentists, pediatric dentists and orthodontists in Belgium. The research explored respondents’ personal experience with patients who had dental trauma in the past and the non-specified management of traumatized teeth. A survey in United Kingdom was conducted by C. Sandler et al. [7] and research by P. M. Tondelli et al. [14] was carried out in Brazil. The aim of the research was to assess the knowledge of orthodontists in the orthodontic management of traumatized teeth.

According to the present survey, the rate of correct answers was 57.4%. A study conducted in Brazil showed that 61.3% of respondents chose the correct answer [14], compared to only 36.5% in the United Kingdom [7]. Such numbers are insufficient and may be due to the great lack of information about dental injuries, so it is very important to organize additional training regarding this topic.

Considering the appropriate medical history is an important factor in the success of treatment. Since only 50% of respondents ask patients about past injuries before treatment, and only 44.4% of patients fill out a special questionnaire before treatment that includes questions about dental trauma, it can be assumed that orthodontists often do not even know they are treating patients who have experienced dental trauma in the past. The study found that respondents were most likely to find out about dental trauma when they noticed signs or complications of a previous dental injury (53.7%), compared to the study from the United Kingdom reporting that 61% of respondents ask about patients’ previous dental trauma before starting orthodontic treatment [7].

When asked about the definition of a concussion, 92.6% of respondents gave the correct answer. The least harmful orthodontic movement to traumatized teeth is extrusion, which produces little biological stimulation in the absence or minimal compression of the periodontal ligament [14]. According to a survey in Brazil [14], only 53.3% answered this question correctly. Lithuanian orthodontists were almost twice as likely to choose the correct answer to the question about the least harmful orthodontic movement for traumatized teeth than Brazilian participants were (38.9% and 20%, respectively) [14].

Evaluating the answers about the recommended observation period before orthodontic treatment, it can be seen that the opinion of Lithuanian orthodontists is very similar to the choices of orthodontists in other countries. About 30% of professionals in Lithuania choose to wait 3 months before starting orthodontic treatment for a tooth experiencing a crown or crown–root fracture. In the case of crown fracture, the second-most popular answer is that orthodontic treatment can be performed immediately. In the case of crown–root fracture, the most common response is a 12-month follow-up period. This indicates that respondents, without knowing the exact answer, choose a longer waiting period depending on the extent of the trauma.

Sandler and co-authors [7] state that it is sufficient to observe a tooth for 6 months after a severe periodontal injury. However, Kindelan et al. [3] suggest that teeth experiencing severe damage to the periodontium should be monitored for up to one year prior to orthodontic treatment, to reduce the likelihood of ankylosis. According to Sönmez H and co-authors, mild orthodontic forces should be used on teeth that have suffered intrusive dental trauma to prevent external root resorption [17].

Sometimes, if a tooth is suspected of ankylosing after trauma, early orthodontic alignment can be performed to prevent the tooth from ankylosing in a disadvantageous position, but this would increase the risk of root shortening during treatment. It is important to inform the patient about the possible risk of root resorption and a worsened long-term prognosis of the tooth. These teeth must be closely monitored regularly throughout orthodontic treatment [6]. This issue is debatable and depends on the severity of the injury. In Lithuania, half of the respondents choose a 12-month observation period. It can be concluded that participants tend to leave the traumatized tooth untreated for longer than recommended to avoid further complications. After minor damage to the periodontium, 64.8% of respondents in Lithuania waited 3 months, compared to 62.9% of respondents in the United Kingdom [7] and only 36.2% of respondents in Brazil [14].

According to Kindelan and co-authors [3], the period of observation of traumatized teeth prior to orthodontic treatment depends on the origin of the trauma, but it is recommended that the patient waits 12 months for endodontic treatment to be successful. Sometimes, depending on the severity of the trauma, orthodontic movements should only be started once the periodontal ligament is healed, infection is under control and stable results are achieved [6]. Only 7.4% of Lithuanian physicians would wait 12 months before starting orthodontic treatment after the endodontic treatment of a traumatized tooth. The low response rate could be due to the question not detailing the severity of the injury or respondents thinking that the question asked about an endodontic treatment due to caries, not due to trauma. In the case of a tooth needing an endodontic treatment due to caries, it is not necessary to observe a tooth before orthodontic treatment [3]. In the United Kingdom, the correct answer response rate was twice as high (14.3%) [7]. In Lithuania, only 20.4% of doctors reported choosing to wait 12 months before orthodontic treatment after regenerative endodontic treatment (RET) due to dental injury, compared to 25.2% of doctors in the United Kingdom [7]. Sandler et al. [6] state that such teeth should be observed for up to 2 years, as internal root resorption or the loss of newly formed pulp tissue may develop in the future. Elfrink and co-authors [18] also suggest that dental ankylosis may occur 18 months after RET. Due to the small number of studies related to regenerative treatment after dental trauma, it is difficult to name the optimal follow-up time [7]. The low number of correct answers from respondents when questions are closely related to endodontic treatment could be due to the fact that such teeth often require assessment by an interdisciplinary team.

Owtad and co-authors [4] state that it is important to assess the condition of the tissues surrounding the root apex frequently (usually every 3 months) using a periapical radiograph during orthodontic treatment. Briefly, 46.3% of the respondents of this study said that they regularly take X-rays of traumatized teeth, and in Brazil a much higher proportion of doctors chose this answer—97.1% [14].

Furthermore, 77.8% of the participants in this study answered that moderate orthodontic forces promote bone repair processes. Orthodontic forces stimulate the periodontal ligament and alveolar bone cells that secrete anti-inflammatory, angiogenic, and osteogenic substances. This triggers the remodeling process of the periodontal ligament and adjacent alveolar bone [19]. Forces created during orthodontic treatment can cause complications to traumatized teeth which will later affect the effects of orthodontic treatment [20]. Immediate referral to a dentist, precise diagnosis, treatment planning and regular follow-up improve long-term prognosis and the positive outcome of the injured tooth [21]. During the organization of the study, it was relevant to find out whether or not the knowledge of orthodontists and orthodontic residents about dental injuries and their orthodontic treatment is sufficient. However, the response rate was only 42.9%; therefore, the collected data cannot represent the knowledge of a country, but the results show the high requirement for further education. There are also no official guidelines for orthodontists regarding the treatment of traumatized teeth. We need to encourage interdisciplinary collaboration and promote education.

During the study, difficulties were encountered in interviewing orthodontists. Although questionnaires were distributed at conferences, published on a website and sent to clinics employing orthodontists, only 42 out of 114 (36.8) responded to the questionnaire, in the UK this percentage was even lower at only 14% [7]. During the orthodontist conferences, the activity of the participants was the highest but due to the pandemic, their number was limited. For future research, we suggest mainly attending and collecting data at orthodontist conferences.

## 5. Conclusions

The knowledge of Lithuanian orthodontists and orthodontic residents about dental injuries and the orthodontic treatment of traumatized teeth is insufficient. Further education, training and interdisciplinary mediation is recommended.

## Figures and Tables

**Figure 1 medicina-59-01289-f001:**
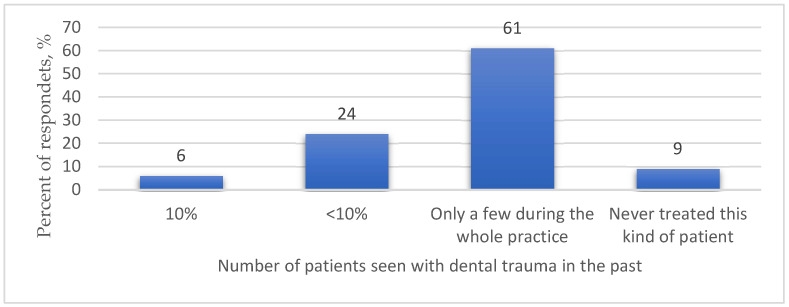
Respondents’ exposure to dental trauma patients in the past.

**Figure 2 medicina-59-01289-f002:**
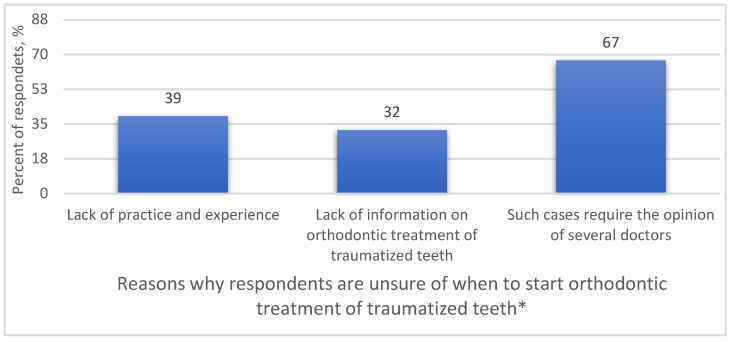
Reasons why physicians do not know when to initiate orthodontic treatment for traumatized teeth. * More than one answer was available.

**Table 1 medicina-59-01289-t001:** Evaluation of physicians’ responses to a question about the way dental trauma was learnt about in the past.

Do You Ask Patients about Past Dental Trauma When Collecting a Medical History? *	In Total	
	*n*	%
I always ask the patient if he had a dental trauma in the past before the orthodontic treatment	27	50
If you notice signs of a previous dental injury, complications	29	53.7
If you notice increased horizontal overjet	3	5.6
The patient completes a special questionnaire before treatment, which also includes questions about dental injuries	24	44.4

* More than one answer was available.

**Table 2 medicina-59-01289-t002:** Evaluation of physicians’ responses to questions about dental trauma.

Questions about Dental Trauma (Correct Answer)	Total Number of Correct Answers of Respondents	
	*n*	%
A displacement of the tooth out of its socket in an incisal/axial direction when there’s no alveolar bone, is classified as: (extrusion)	32	59.3
An injury to the tooth supporting structures without increasing tooth mobility or displacement but with significant sensibility to the percurssion, is classified as: (concussion)	50	92.6
Which device would you choose during the period of acute dental trauma (the patient arrived immediately after the injury, changes in the position of the tooth, damage to the periodontal ligament)? (Segmental bracket-system)	22	40.7
At what age do dental injuries occur the most? (7–14 years)	32	59.3
What recovery method would you choose for a tooth that has been submerged in the alveolus during an injury? (Surgical extrusion and immobilization or orthodontic extrusion immediately after injury)	36	66.7

**Table 3 medicina-59-01289-t003:** Evaluation of physicians’ responses to questions about orthodontic treatment of traumatized teeth.

Questions about Orthodontic Management of Traumatized Teeth (Correct Answer)	Total Number of Correct Answers of Respondents	
	*n*	%
In your opinion, which orthodontic movement is the least harmful to teeth who have suffered an injury? (Extrusion)	21	38.9
How long is it estimated to wait prior to orthodontic treatment after a crown fracture? (3 months)	20	37
How long is it estimated to wait prior to orthodontic treatment after a crown-root fracture? (3 months)	12	22.2
How long is it estimated to wait prior to orthodontic treatment after a minor damage to the periodontum? (3–6 months)	35	64.8
How long is it estimated to wait prior to orthodontic treatment after a severe damage to the periodontum? (6 months)	19	35.2
How long is it estimated to wait prior to orthodontic treatment for a tooth requiring endodontic treatment due to trauma? (12 months)	4	7.4
How long is it estimated to wait prior to orthodontic treatment for a tooth requiring regenerative endodontic treatment due to trauma? (12 months)	11	20.4
Do you take X-rays regularly (every 3 months) during orthodontic treatment of traumatized teeth? (Yes)	25	46.3
In your opinion, can a tooth that has suffered a root fracture in the past be subjected to normal orthodontic forces? (No)	35	64.8
In your opinion, can an injured tooth experience greater (compared to intact teeth) root resorption due to orthodontic forces? (Yes)	52	96.3
Do you think there is a possibility that other teeth have been injured during the trauma, but the consequences of the trauma are not visible until orthodontic treatment is initiated? (Yes)	48	88.9
How would you react if you notice a change in the color of a patient’s tooth during orthodontic treatment? (I would eliminate the action of orthodontic forces directed towards the damaged tooth until the cause of such changes is resolved)	47	87
How long is it estimated to wait prior to orthodontic treatment for a tooth with signs of inflammatory resorption? (When complete healing and integrity of periodontal ligament are observed and regular (every 3 months) X-ray examination is taken for one year)	51	94.4
Do you think that rational orthodontic forces promote bone repair processes? (Yes)	42	77.8

## Data Availability

Data are contained within the article.

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
