# Peer review of "Knowledge of Dental Trauma and Orthodontic Management of Traumatized Teeth by a Group of Lithuanian Orthodontists"

_medicina, 2023, doi:10.3390/medicina59071289_

Round 1

Reviewer 1 Report

Dear Authors, 

Thank you for your submission to Medicina. 

Here are a few points to improve the manuscript for publication, please note that many limitations of the study are present and should be discussed:

- the study sample is limited if you consider that this is a survey study

- no statistical analysis was performed as only dicotomic variables were recorded 

- no future perspectives are present 

- the discussion should be implemented with suggested ways to improve the lack of knowledge from orthodontists and orthodontic residents. 

Minor English corrections should be performed.

Author Response

Dear Reviewer,

Thank you for your comments and suggestions.

We will try to answer your comments as accurately as possible.

The study sample is limited if you consider that this is a survey study

The sample is limited because there are a small number of orthodontists in Lithuania and some of them refused to take part in the survey or did not complete the questionnaire, possibly due to the complexity of the questionnaire or lack of knowledge.

No statistical analysis was performed as only dicotomic variables were recorded

The survey was developed based on 3 other studies. 2 of them did not have statistical analysis and the other compared 3 groups of specialists (orthodontists, general dentists and pediatric dentists)

No future perspectives are present 

We respect your opinion about the prospects of the study. We just want to add that our goal is to draw attention to the lack of knowledge on this topic and to encourage interdisciplinary cooperation to ensure education.

The discussion should be implemented with suggested ways to improve the lack of knowledge from orthodontists and orthodontic residents.

We have revised the discussion based on your comments.

We hope we answered your questions and comments in detail. Please do not hesitate and share with us how to improve our article even more.

Reviewer 2 Report

Dear authors, 

Please review the methods and the discission. See the document enclosed as an attachment

Author Response

Dear Reviewer,

Thank you for your comments and suggestions.

We will try to answer your comments as accurately as possible.

Lack information about handling of the data, Descriptive statistics?

We have added comments on data collection and analysis.

The frequency of correct answers to the survey is 57.1% for practicing orthodontists and 58.3% for orthodontic residents (total 57.4%). There were no significant statistical differences between these two groups of specialists, therefore the data were not further analyzed and described in detail.

We hope we answered your questions and comments in detail. Please do not hesitate and share with us how to improve our article even more.

Reviewer 3 Report

It is an interesting study, about the level of knowledge about the treatment of traumatized teeth of Lithuanian practitioners.

For enhancing the strength of the manuscript few aspects should be addressed:

Introduction:

          -the results of similar studies about the knowledge of orthodontics should be added

           -the programs, measures taken to address these issues (if they exist)

            -a more detailed description about the orthodontic traumatism that are more likely to be found in clinical practice

Material and methods:

         -the number of ethical committee acceptance report should be mentioned also in this chapter

         -a few statistical data: sample size (it seems to be little for such subject- thus the motivation that it is trustworthy), relevance, criteria of inclusion, exclusion, the total number of participants (the number of academics, private practitioners, and interns, will also be of interest). Despite some of these being found in the results and discussion chapters, they should be briefly mentioned in the methodology.

Author Response

Dear Reviewer,

Thank you for your comments and suggestions.

We will try to answer your comments as accurately as possible.

We have revised the introduction based on your comments.

           -the programs, measures taken to address these issues (if they exist)

All measures are of a recommendatory nature, but the continuous publicity of the problem has led to personal development of orthodontists and to the improvement of the curriculum for future orthodontists.

            -a more detailed description about the orthodontic traumatism that are more likely to be found in clinical practice 

We added this in to introduction.

Material and methods:

         -the number of ethical committee acceptance report should be mentioned also in this chapter

We added this information.

         -a few statistical data: sample size (it seems to be little for such subject- thus the motivation that it is trustworthy), relevance, criteria of inclusion, exclusion, the total number of participants (the number of academics, private practitioners, and interns, will also be of interest). Despite some of these being found in the results and discussion chapters, they should be briefly mentioned in the methodology.

We added this information.

We hope we answered your questions and comments in detail. Please do not hesitate and share with us how to improve our article even more.

Reviewer 4 Report

The comments are included in the file below.

Minor revision related to the quality of English.

Author Response

Dear Reviewer,

Thank you for your comments and suggestions.

We will try to answer your comments as accurately as possible.

We have added the ethical committe acceptance report and information about the data collection and analysis.

The case control citation has been corrected in the discussion.

In discussion we compared with similar studies in other countries. In the United Kingdom and Brazil respondents were specialists in orthodontics.

We hope we answered your questions and comments in detail. Please do not hesitate and share with us how to improve our article even more.

Round 2

Reviewer 2 Report

no comments

Reviewer 4 Report

Dear Authors,

Thanks for providing the answers I needed to better assess your manuscript.

Best Regards,

Dear Authors,

Please check the whole text once more to be sure your English is fine.

Best Regards,